# Representation Learning Dynamics of Self-Supervised Models

**Pascal M. Esser**$^*$                                                                    *esser@cit.tum.de*
*Technical University of Munich, Germany*

**Satyaki Mukherjee**$^*$                                                          *paglasatyaki@gmail.com*
*National University of Singapore, Singapore*

**Debarghya Ghoshdastidar**                                                          *ghoshdas@cit.tum.de*
*Technical University of Munich, Germany*

**Reviewed on OpenReview:** *https://openreview.net/forum?id=QXLKnrymE1*

## Abstract

Self-Supervised Learning (SSL) is an important paradigm for learning representations from unlabelled data, and SSL with neural networks has been highly successful in practice. However current theoretical analysis of SSL is mostly restricted to generalisation error bounds. In contrast, learning dynamics often provide a precise characterisation of the behaviour of neural networks based models but, so far, are mainly known in supervised settings. In this paper, we study the learning dynamics of SSL models, specifically representations obtained by minimising contrastive and non-contrastive losses. We show that a näive extension of the dynamics of multivariate regression to SSL leads to learning trivial scalar representations that demonstrates dimension collapse in SSL. Consequently, we formulate SSL objectives with orthogonality constraints on the weights, and derive the exact (network width independent) learning dynamics of the SSL models trained using gradient descent on the Grassmannian manifold. We also argue that the infinite width approximation of SSL models significantly deviate from the neural tangent kernel approximations of supervised models. We numerically illustrate the validity of our theoretical findings, and discuss how the presented results provide a framework for further theoretical analysis of contrastive and non-contrastive SSL.

## 1 Introduction

A common way to distinguish between learning approaches is to categorize them into unsupervised learning, which relies on a input data consisting of a feature vector $(x)$, and supervised learning which relies on feature vectors and corresponding labels $(x, y)$. However, in recent years, Self-Supervised Learning (SSL) has been established as an important paradigm between supervised and unsupervised learning as it does not require explicit labels but relies on implicit knowledge of what makes some samples semantically close to others. Therefore SSL builds on inputs and inter-sample relations $(x, x^+)$, where $x^+$ is often constructed through data-augmentations of $x$ known to preserve input semantics such as additive noise or horizontal flip for an image Kanazawa et al. (2016); Novotny et al. (2018); Gidaris et al. (2018). While the idea of SSL is not new Bromley et al. (1993), recent deep SSL models have been highly successful in computer vision Chen et al. (2020); Caron et al. (2021); Jing & Tian (2019), natural language processing Misra & Maaten (2020); Devlin et al. (2019), speech recognition Steffen et al. (2019); Mohamed et al. (2022). Since the early works Bromley et al. (1993), methods for SSL have predominantly relied on neural networks however with a strong focus on model design with only little theoretical backing.

---

$^*$P. M. Esser and S. Mukherjee equally contributed to this work. The work was carried out while S. Mukherjee was at the Technical University of Munich.

While there are theoretical works addressing SSL specific question such as Wen & Li (2021) analyzing the importance of data augmentation and Pokle et al. (2022) characterizing 'dimension collapse' through the loss landscape, the main focus of the theory literature on SSL has been either on providing generalization error bounds for downstream tasks on embeddings obtained by SSL (Arora et al., 2019b; Ge et al., 2023; Bansal et al., 2021; Lee et al., 2021; Saunshi et al., 2021; Tosh et al., 2021; Wei et al., 2021; Bao et al., 2022; Chen et al., 2022), or analysing the spectral and isoperimetric properties of data augmentation (Balestriero & LeCun, 2022; Han et al., 2023; Zhuo et al., 2023). The latter approach also result in novel bounds on the generalisation error (HaoChen et al., 2021; Zhai et al., 2023). While generalisation theory remains one of the fundamental tools to characterise the statistical performance, it has been already established for supervised learning that classical generalisation error bounds do not provide a complete theoretical understanding and can become trivial in the context of neural network models (Zhang et al., 2017; Neyshabur et al., 2017). Therefore a key focus in modern deep learning theory is to understand the learning dynamics of models, often under gradient descent, as they provide a more tractable expression of the problem that can be an essential tool to understand the loss landscape and convergence (Fukumizu, 1998; Saxe et al., 2014; Pretorius et al., 2018), early stopping (Li et al., 2021), linearised (kernel) approximations (Jacot et al., 2018; Du et al., 2019) and, mostly importantly, generalisation and inductive biases (Soudry et al., 2018; Luo et al., 2019; Heckel & Yilmaz, 2021).

*In this paper, we analyze the learning dynamics of SSL models under contrastive and non-contrastive losses (Arora et al., 2019b; Chen et al., 2020), which we show to be significantly different from the dynamics of supervised models.* This gives a simple and precise characterization of the dynamics that can provide the foundation for future theoretical analysis of SSL models.

**Contrastive Learning.** Contrastive SSL has its roots in the work of Bromley et al. (1993). Recent deep learning based contrastive SSL show great empirical success in computer vision (Chen et al., 2020; Caron et al., 2021; Jing & Tian, 2019), video data (Fernando et al., 2017; Sermanet et al., 2018), natural language tasks (Misra & Maaten, 2020; Devlin et al., 2019) and speech (Steffen et al., 2019; Mohamed et al., 2022). In general a contrastive loss is defined by considering an anchor image, $x \in \mathbb{R}^d$, positive samples $\{x^+\} \subset \mathbb{R}^d$ generated using data augmentation techniques as well as independent negative samples $\{x^-\} \subset \mathbb{R}^d$. The heuristic goal is to align the anchor more with the positive samples than the negative ones, which is rooted in the idea of maximizing mutual information between similar samples of the data. In this work, we consider a simple contrastive loss minimisation problem along the lines of Arora et al. (2019b), assuming exactly one positive sample $x_i^+$ and one negative sample $x_i^-$ for each anchor $x_i$,[1]

$$\min_{\Theta} \sum_{i=1}^n u(x_i)^\top \left( u(x_i^-) - u(x_i^+) \right), \tag{1}$$

where $u = [u_1(\cdot, \Theta) \dots u_z(\cdot, \Theta)]^\top : \mathbb{R}^d \to \mathbb{R}^z$ is the embedding function, parameterized by $\Theta$.

**Contributions.** The objective of this paper is to derive the evolution dynamics of the learned embedding $u = u(\cdot, \theta)$ under gradient flow for the contrastive (1). More specifically we show the following with proofs provided in the supplementary material:

- We express the learning dynamics contrastive learning and show that, the evolution dynamics is same across dimensions. This explains why SSL is naturally prone to dimension collapse.

- Assuming a 2-layer linear network, we show that dimension collapse cannot be avoided by adding standard Frobenius norm regularisation, but by adding orthogonality or L2 norm constraints.

- We further show that at initialization, the dynamics of 2-layer network with nonlinear activation is close to their linear, width independent counterparts (Theorem 1). We also provide empirical evidence that the evolution of the infinite width non-linear networks are close to their linear counterparts, under certain conditions on the non-linearity (that hold for tanh).

---

[1]It is straightforward to extend our analysis to multiple positive and negative samples, but the expressions become cumbersome, without providing additional insights.

- Going beyond the 2-layer setting, we derive the learning dynamics of SSL for deep linear networks, under orthogonality constraints (Theorem 2). We further show the convergence of the learning dynamics for the one dimensional embeddings ($z = 1$).

- We numerically show, on the MNIST dataset, that our derived SSL learning dynamics can be solved significantly faster than training nonlinear networks, and yet provide comparable accuracy on downstream tasks.

- Finally we show that the presented results extend to simple non-contrastive losses as well.

**Related works.** Our focus is on the evolution of the learned representations, and hence, considerably different from the aforementioned literature on generalisation theory and spectral analysis of SSL. From an optimisation perspective, Liu et al. (2023) derive the loss landscape of contrastive SSL with linear models, $u(x) = Wx$, under InfoNCE loss van den Oord et al. (2018). Although the contrastive loss in (1) seems simpler than InfoNCE, they are structurally similar under linear models (Liu et al., 2023, see Eqns. 4–6). Similarly Zbontar et al. (2021) maximizes the variability of the embeddings by decorrelating the components of the embeddings vectors. Training dynamics for contrastive SSL with linear shallow models (with only one hidden layer , $u(x) = Wx$) have been derived in Simon et al. (2023) under a simplified Barlow twins loss and deep linear models have been partially investigated by Tian (2022), who show an equivalence with principal component analysis, and by Jing et al. (2022), who establish that dimension collapse occurs for over-parametrised linear contrastive models. Theorem 2 provides a more precise characterisation and convergence criterion of the evolution dynamics than previous works. Furthermore, none of prior works consider non-linear models or orthogonality constraints as studied in this work.

We also distinguish our contributions (and discussions on neural tangent kernel connections) with the kernel equivalents of SSL studied in Kiani et al. (2022); Johnson et al. (2023); Shah et al. (2022); Cabannes et al. (2023). While Shah et al. (2022); Cabannes et al. (2023) specifically pose SSL objectives using kernel models, Kiani et al. (2022); Johnson et al. (2023) show that contrastive SSL objectives induce specific kernels. Importantly, these works neither study the learning dynamics nor consider the neural tangent kernel regime.

**Towards analysing deep, non-linear networks.** Analyzing deep, non-linear networks presents a complex challenge and no unifying approach has been established even in the more studied supervised setting. Existing works usually consider one (or several) of the following assumptions: *(a) linear networks.* This allows for an exact characterization of the network (Saxe et al., 2014; Ziyin et al., 2022; Basu et al., 2019) however the considered proof techniques do not extend to the non-linear setting. *(b) strong data / initialization assumptions.* Tachet et al. (2018); Mei & Montanari (2022) are able to derive exact solutions in the non-linear setting but the proof structure breaks down if the assumptions (which might not be used in practice) are violated. *(c) strong architecture assumptions.* Jacot et al. (2018); Arora et al. (2019b) derive dynamics for deep, non-linear networks, however need strong assumptions on the initialization and the (infinite) width of the network. How exactly the behaviour of finite and infinite wide networks relate is still an open question.

In this work we aim to analyze settings that are relevant in practice but still allow for a exact theoretical analysis. We address the challenge of analyzing non-linearities and depth by first showing that linear and non-linear networks are close in Theorem 1 and secondly by deriving dynamics for deep linear networks in Theorem 2. In addition our results only have mild initialisation (orthogonality) and data assumptions.

**Notation.** Let $\mathbb{I}_n$ be an $n \times n$ identity matrix. For a matrix $A$ let $\|A\|_F$ and $\|A\|_2$ be the standard frobenious norm and the $L2$-operator norm respectively. The machine output is denoted by $u(\cdot)$. While $u$ is time dependent and should be more accurately denoted as $u_t$ we suppress the subscript where obvious. For any time dependent function, for instance $u$, denote $\mathring{u}$ to be its time derivative i.e. $\frac{du_t}{dt}$. $\phi$ is used to denote our non-linear activation function and we abuse notation to also denote its co-ordinate-wise application on a vector by $\phi(\cdot)$. $\langle \cdot, \cdot \rangle$ is used to denote the standard dot product.

## 2 Learning Dynamics of Regression and its Näive Extension to SSL

In the context of regression, Jacot et al. (2018) show that the evolution dynamics of (infinite width) neural networks, trained using gradient descent under a squared loss, is equivalent to that of specific kernel machines,

known as the neural tangent kernels (NTK). The analysis has been extended to a wide range of models, including convolutional networks (Arora et al., 2019a), recurrent networks (Alemohammad et al., 2021), overparametrised autoencoders (Nguyen et al., 2021), graph neural networks (Du et al., 2019; Sabanayagam et al., 2022) among others. However, these works are mostly restricted to squared losses, with few results for margin loss (Chen et al., 2021), but derivation of such kernel machines are still open for contrastive or non-contrastive losses, or broadly, in the context of SSL. To illustrate the differences between regression and SSL, we outline the learning dynamics of multivariate regression with squared loss, and discuss how a näive extension to SSL is inadequate.

## 2.1 Learning Dynamics of Multivariate Regression

Given a training feature matrix $X := [x_1, \cdots, x_n]^\top \in \mathbb{R}^{n \times d}$ and corresponding $z$-dimensional labels $Y := [y_1, \cdots, y_n]^\top \in \mathbb{R}^{n \times z}$, consider the regression problem of learning a neural network function $u(x) = [u_1(x, \Theta) \ldots u_z(x, \Theta)]^\top$, parameterized by $\Theta$, by minimising the squared loss function $\mathcal{L}(\Theta) := \frac{1}{2} \sum_{i=1}^n \|u(x_i) - y_i\|^2$. Under gradient flow, the evolution dynamics of the parameter during training is $\mathring{\Theta} = -\nabla_\Theta \mathcal{L}$ and, consequently, the evolution of the $l$-th component of network output $u(x)$, for any input $x$, follows the differential equation

$$\mathring{u}_l(x) = \left\langle \nabla_\Theta u_l(x), \mathring{\Theta} \right\rangle = -\sum_{i=1}^n \sum_{j=1}^z \langle \nabla_\Theta u_l(x), \nabla_\Theta u_j(x_i) \rangle \left( u_j(x_i) - y_{i,j} \right). \tag{2}$$

While the above dynamics apparently involve interaction between the different dimensions of the output $u(x)$, through $\langle \nabla_\Theta u_l(x), \nabla_\Theta u_j(x_i) \rangle$, it is easy to observe that this interaction does not contribute to the dynamics of linear or kernel models. We formalise this in the following lemma.

**Lemma 1 (No interaction across output dimensions).** *Let $u : \mathbb{R}^d \to \mathbb{R}^z$ be either a linear model $u(x) = \Theta x$, or a kernel machine $u(x) = \Theta \psi(x)$, where $\psi$ corresponds to the implicit feature map of a kernel $k$, that is, $k(x, x') = \langle \psi(x), \psi(x') \rangle$.*
*Then in the infinite width limit ($h \to \infty$) the inner products between the gradients are given by*

$$\langle \nabla_\Theta u_l(x), \nabla_\Theta u_j(x') \rangle = \begin{cases} 0 & \text{if } l \neq j, \\ x^\top x' & \text{if } l = j \text{ (linear case)}, \\ k(x, x') & \text{if } l = j \text{ (kernel case)}. \end{cases}$$

For infinite width neural networks, whose weights are randomly initialised with appropriate scaling, Jacot et al. (2018) show that at, initialisation, Lemma 1 holds with $k$ being the neural tangent kernel. Approximations for wide neural networks further imply the kernel remains same during training (Liu et al., 2020), and so Lemma 1 continues to hold through training.

**Remark 1 (Multivariate regression = independent univariate regressions).** *A consequence of Lemma 1 is that the learning dynamics (2) simplifies to*

$$\mathring{u}_l(x) = -\sum_{i=1}^n \langle \nabla_\Theta u_l(x), \nabla_\Theta u_l(x_i) \rangle \left( u_l(x_i) - y_{i,l} \right),$$

*that is, each component of the output $u_l$ evolves independently from other $u_j, j \neq l$. Hence, one may solve a $z$-variate squared regression problem as $z$ independent univariate problems. We discuss below that a similar phenomenon is true in SSL dynamics with disastrous consequences.*

## 2.2 Dynamics of Näive SSL has a Trivial Solution

We now present the learning dynamics of SSL with contrastive losses (1). Assuming that the network function $u : \mathbb{R}^d \to \mathbb{R}^z$ is parameterised by $\Theta$, the gradient of the loss $\mathcal{L}(\Theta) = \sum_{i=1}^n u(x_i)^\top \left( u(x_i^-) - u(x_i^+) \right)$ is

$$\nabla_\Theta \mathcal{L}(\Theta) = -\sum_{i=1}^n \sum_{j=1}^z u_j(x_i) \cdot \nabla_\Theta u_j(x_i^+) + u_j(x_i^+) \cdot \nabla_\Theta u_j(x_i) - u_j(x_i) \cdot \nabla_\Theta u_j(x_i^-) - u_j(x_i^-) \cdot \nabla_\Theta u_j(x_i)$$

Hence, under gradient descent $\mathring{\Theta} = -\nabla_\Theta \mathcal{L}$, the evolution of each component of $u(x)$, given by $\mathring{u}_l(x) = \left\langle \nabla_\Theta u_l(x), \mathring{\Theta} \right\rangle$ is expressed by

$$\mathring{u}_l(x) = \sum_{i=1}^n \sum_{j=1}^z \langle \nabla_\Theta u_l(x), \nabla_\Theta u_j(x_i) \rangle \, u_j(x_i^+) + \langle \nabla_\Theta u_l(x), \nabla_\Theta u_j(x_i^+) \rangle \, u_j(x_i)$$
$$- \langle \nabla_\Theta u_l(x), \nabla_\Theta u_j(x_i) \rangle \, u_j(x_i^-) - \langle \nabla_\Theta u_l(x), \nabla_\Theta u_j(x_i^-) \rangle \, u_j(x_i). \tag{3}$$

We note Lemma 1 depends only on the model and not the loss function, and hence, it is applicable for the SSL dynamics in (3). However, there are no multivariate training labels $y \in \mathbb{R}^z$ in SSL (i.e. $y = 0$) that can drive the dynamics of the different components $u_1, \ldots, u_z$ in different directions, which leads to dimension collapse.

**Proposition 1** (**Dimension collapse in SSL dynamcis**). *Under the conditions of Lemma 1, every component of the network output $u : \mathbb{R}^d \to \mathbb{R}^z$ has identical dynamics. As a consequence, the output collapses to one dimension at convergence. For linear model, $u(x) = \Theta x$, the dynamics of $u(x)$ is given by*

$$\mathring{u}_l(x) = \sum_{i=1}^n (x^\top x_i)\big(u_l(x_i^+) - u_l(x_i^-)\big) + (x^\top x_i^+ - x^\top x_i^-)u_l(x_i)$$

*for a contrastive model. For kernel models, the dynamcis is similarly obtained by replacing each $x^\top x'$ by $k(x, x')$.*

By the extension of Lemma 1 to neural network and NTK dynamics, one can conclude that Proposition 1 and dimension collapse also happen for wide neural networks, when trained for the contrastive loss (1).

**Remark 2** (**SSL dynamics for other losses**). *One may argue that the above dimension collapse is a consequence of loss definition in (1), and may not exist for other losses. We note that Liu et al. (2023) analyse contrastive learning with linear model under InfoNCE, and the simplified loss closely resembles (1), which implies decoupling of output dimensions (and hence, dimension collapse) would also happen for InfoNCE. The same argument also holds for non-constrastive loss in Chen et al. (2020). However, for the spectral contrastive loss of HaoChen et al. (2021), the output dimensions remain coupled in the SSL dynamics due to existing interactions $u(x_i)^\top u(x_i^-)$ on the training data.*

**Remark 3** (**Projections cannot overcome dimension collapse**). *Jing et al. (2022) propose to project the representation learned by a SSL model into a much smaller dimension, and show that fixed (non trainable) projectors may suffice. For a linear model, this implies $u(x) = A\Theta x$, where $A \in \mathbb{R}^{r \times z}, r \ll z$ is fixed. It is straightforward to adapt the dynamics and Proposition 1 to this case, and observe that for any $r > 1$, all the $r$ components of $u(x)$ have identical learning dynamics, and hence, collapse at convergence.*

## 3    SSL with (Orthogonality) Constraints

For this section, we assume that the SSL model $u : \mathbb{R}^d \to \mathbb{R}^z$ corresponds to a 2-layer neural network of the form

$$u(x) = W_2^\top \phi(W_1 x) \in \mathbb{R}^z,$$

where $h$ is the size of the hidden layer and $\Theta = (W_1, W_2^\top)$ are trainable matrices. Whenever needed, we use $u^\phi$ for the output to emphasize the nonlinear activation $\phi$, and contrast it with a 2-layer linear network $u^{\mathbb{I}}(x) = W_2^\top W_1 x$. The optimization problem can now be restated as a trace minimization problem which has been previously observed in Balestriero et al. (2023). Assume the linear setting, we can write our loss function (1) as

$$\mathcal{L} = \sum_i^n \text{Tr}\left(W_2^\top W_1 x_i \left(x_i^- - x_i^+\right)^\top W_1^\top W_2\right) = \text{Tr}\left(W_2^\top W_1 \widetilde{C} W_1^\top W_2\right) = \text{Tr}\left(W_2^\top W_1 C W_1^\top W_2\right) \tag{4}$$

with

$$C = \frac{\widetilde{C} + \widetilde{C}^\top}{2} \quad \text{and} \quad \widetilde{C} = \sum_i^n x_i \left( x_i^- - x_i^+ \right)^\top. \tag{5}$$

Furthermore (1) can easily be extended to the $p$ positive and $q$ negative sample setting where we then obtain $\widetilde{C} = \sum_i^n \left( \sum_j^q x_i \left( x_j^- \right)^\top - \sum_j^p x_i \left( x_j^+ \right)^\top \right)$.

Based on the discussion in the previous section, it is natural to ask how can the SSL problem be rephrased to avoid dimension collapse. An obvious approach is to add regularisation or constraints (Bardes et al., 2021; Ermolov et al., 2021; Caron et al., 2020). The most obvious regularisation or constraint on $W_1, W_2$ is entry-wise, such as on Frobenius norm. While there has been little study on various regularisations in SSL literature, a plethora of variants for Frobenius norm regularisations can be found for autoencoders, such as sum-regularsiation, $\|W_1\|_F^2 + \|W_2\|_F^2$, or product regularisation $\|W_2^\top W_1\|_F^2$ (Kunin et al., 2019).

It is known in the optimisation literature that regularised loss minimisation can be equivalently expressed as constrained optimisation problems. In this paper, we use the latter formulation for convenience of the subsequent analysis. The following result shows that Frobenius norm constraints do not prevent the output dimensions from decoupling, and hence, it is still prone to dimension collapse.

**Proposition 2** (**Frobenius norm constraint does not prevent dimension collapse**). *Consider a linear embedding function $u^\phi(x) = W_2^\top \phi(W_1 x)$ and let $\mathcal{L}(W_1, W_2)$ be given by (1), then the optimisation problem*

$$\min_{W_1, W_2} \mathcal{L}(W_1, W_2) \quad s.t. \quad \|W_1\|_F \le c_1, \|W_2\|_F \le c_2,$$

*with constants $c_1$ and $c_2$ has a global solution $u(x) = [u(x)_1 \ 0 \dots 0]^\top \in \mathbb{R}^z$.*

The above result precisely shows dimension collapse for linear networks $u^{\mathbb{I}}$ even with Frobenius norm constraints. Intuitively we note that the Frobenius norm is defined as the sum of the norms of each column. Critically it therefore does not have any control over the inner products between the columns and has no control over the rank. As such there is nothing preventing all the columns from being the same. Therefore we consider to constrain the $L2$-operator norm as na alternative to Frobenius norm constraint. To this end, the following result shows that, for linear networks, the operator norm constraint can be realised in multiple equivalent ways.

**Proposition 3** (**Equivalence of loss under operator norm and orthogonality constraints**). *Consider a linear embedding function $u^{\mathbb{I}}(x) = W_2^\top W_1 x$, and let the loss $\mathcal{L}(W_1, W_2)$ be given by (1) whose general form is given in (4). Let $C$ be defined by (5) and has atleast one negative eigenvalue. Then there exists a $W_1, W_2$ that is optimal for all following optimization problems:*

1. $\displaystyle \min_{W_1, W_2} \frac{\mathcal{L}(W_1, W_2)}{\|W_2\|_2^2 \|W_1\|_2^2}$;

2. $\displaystyle \min_{W_1, W_2} \mathcal{L}(W_1, W_2) \quad s.t. \quad \|W_2\|_2 \le 1, \ \|W_1\|_2 \le 1$;

3. $\displaystyle \min_{W_1, W_2} \mathcal{L}(W_1, W_2) \quad s.t. \quad \|W_2^\top W_1\|_2 \le 1$;

4. $\displaystyle \min_{W_1, W_2} \mathcal{L}(W_1, W_2) \quad s.t. \quad W_2^\top W_2 = \mathbb{I}_z, \ W_1^\top W_1 = \mathbb{I}_d$.

Avoidance of dimensional collapse is also heuristically evident in the orthogonality constraint $W_2^\top W_2 = \mathbb{I}_z, \ W_1^\top W_1 = \mathbb{I}_d$, which we focus on in the subsequent sections. In particular we observe from the proof of Prop 3 that this regularization extracts the eigenvectors of $C$ corresponding to its "most-negative" eigenvalues

**Example 1** (**SSL dynamics on half moons**). *We numerically illustrate the importance of constraints in SSL. We consider a contrastive setting with the loss in (1) and $u(x) = W_2^\top \tanh(W_1 x)$ for the dataset shown in Figure 1a, where $x^-$ is an independent sample from the dataset and $x^+ = x + \varepsilon$ where $\varepsilon \sim \mathcal{N}(0, 0.1\mathbb{I})$. Let us now compare the dynamics of $\mathcal{L}$ (no constraints) and $\mathcal{L}_{orth}$, the scaling loss that corresponds to orthogonality*

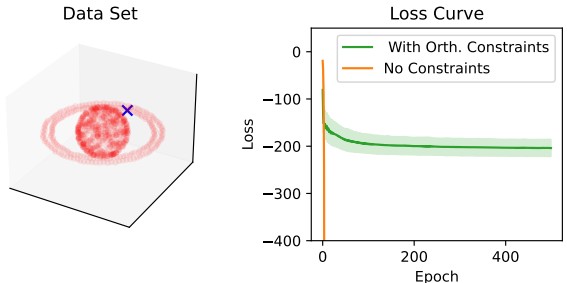

(a) Dataset and loss. **(left)** Illustration of the dataset in $\mathbb{R}^3$. The considered test point is marked with the blue cross. **(right)** Loss curve (mean over 100 initializations) for the network with and without orthogonal constraint.

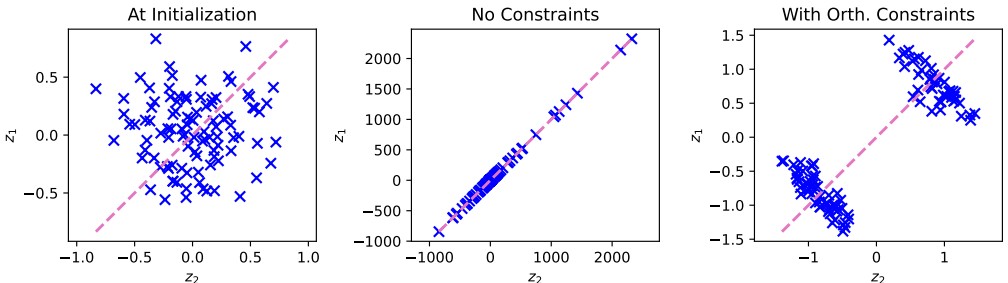

(b) Embedding into $\mathbb{R}^2$. **(left)** Embedding at initialization. **(middle)** Embedding for the network without constraints after 30 epochs. **(right)** Embedding for the network with orthogonal constraints after 500 epochs.

Figure 1: Illustration of dimension collapse. We consider the Dataset in Figure 1a and plot the loss for constraint and unconstrained models. In Figure 1b we furthermore plot the embedding at initialization and during training for both models.

*constraints, and present the results in Figure 1. We firstly observe in Figure 1a that the unconstrained loss does not converge. Secondly considering the embeddings as plotted in Figure 1b we observe dimension collapse for the unconstrained loss function (middle) but not for the one with orthogonal constraints (right).*

### 3.1 Non-Linear SSL Models are Almost Linear

While the above discussion pertains to only linear models, we now show that the network, with nonlinear activation $\phi$ and orthognality constraints,

$$u^\phi_{(t)}(x) = W_2^\top \phi(W_1 x) \quad \text{s.t.} \quad W_2^\top W_2 = \mathbb{I}_z, \ W_1^\top W_1 = \mathbb{I}_d,$$

is almost linear. For this discussion, we explicitly mention the time dependence as a subscript $u^\phi_{(t)}$. We begin by arguing theoretically that in the infinite width limit at initialization there is very little difference between the output of the non-linear machine $u^\phi_{(0)}$ and that of its linear counterpart $u^\mathbb{I}_{(0)}$.

**Theorem 1** (**Comparison of Linear and Non-linear Network**). *Recall that $u_{(t)}$ provides the output of the machine at time t and therefore consider the linear and non-linear setting at initialization as*

$$
\begin{aligned}
u^\mathbb{I}_{(0)} &= W_2^\top W_1 x & \text{s.t.} \quad W_2^\top W_2 = \mathbb{I}_z, \ W_1^\top W_1 = \mathbb{I}_d; \\
u^\phi_{(0)} &= W_2^\top \phi(W_1 x) & \text{s.t.} \quad W_2^\top W_2 = \mathbb{I}_z, \ W_1^\top W_1 = \mathbb{I}_d.
\end{aligned}
\tag{6}
$$

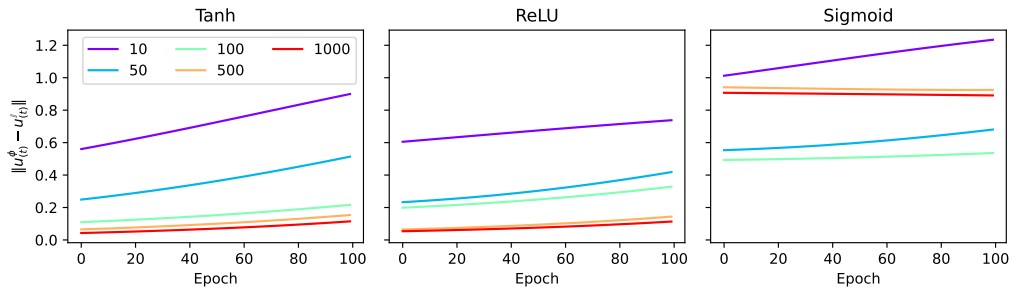

Figure 2: Difference between the non-linear output and the linear output under various conditions on the activation function. Change of the difference while training for hidden layer size 10 to 1000.

*Let $\phi(\cdot)$ be an activation function, such that $\phi(0) = 0$, $\phi'(0) = 1$, and $|\phi''(\cdot)| \leq c$. [2] Then at initialization as uniformly random orthogonal matrices*

$$\left\| u_{(0)}^{\phi} - u_{(0)}^{\mathbb{I}} \right\| \leq Kc\, \|x\|^2\, d\sqrt{\frac{\log^4 h}{h}}$$

*where $K$ is a universal constant $\phi$, $d$ is the feature dimension and $h$ the width of the hidden layer.*

We furthermore conjecture that the same behaviour holds during evolution.

**Conjecture 1** (**Evolution of Non-linear Networks**). *Consider the setup of Theorem 1 with the linear $\left( u_{(t)}^{\mathbb{I}} \right)$ and non-linear machine $\left( u_{(t)}^{\phi} \right)$ as defined in* (6) *and an optimization of the general*

$$\min_{W_2 W_1} \mathrm{Tr}\left( u_{(t)}^{\top} u_{(t)} \right) \quad \text{s.t.} \quad W_2^{\top} W_2 = \mathbb{I}_z, W_1^{\top} W_1 = \mathbb{I}_d.$$

*Again assume $\phi$ is an activation function, such that $\phi(0) = 0$ and $\phi'(0) = 1$. Then*

$$\left\| u_{(t)}^{\phi} - u_{(t)}^{\mathbb{I}} \right\| \to 0 \quad \forall t > 0 \text{ as } h \to \infty.$$

Numerical justification of the above conjecture is presented in the following section.

## 3.2 Numerical Evaluation

We now illustrate the findings of of Theorem 1 and Conjecture 1 numerically. For evaluation we use the following experimental setup: We train a network with contrastive loss as defined in (1) using gradient descent with learning rate 0.01 for 100 epochs and hidden layer size from 10 to 1000. We consider the following three loss functions: (1) sigmoid, (2) ReLU ($\phi(x) = \max\{x, 0\}$) and (3) tanh. The results are shown in Figure 2 where the plot shows the average over 10 initializations. We note that *tanh* fulfills the conditions on $\phi$ and we see that with increasing layer size the difference between linear and non-linear goes to zero. While *ReLU* only fulfills $\phi(0) = 0$ the overall picture still is consistent with *tanh* but with slower convergence. Finally the results on *sigmoid* (which has a linear drift consistent with its value at 0) indicate that the conditions on $\phi$ are necessary as we observe the opposite picture: with increased layer width the difference between linear and non-linear increases.

This overall picture (with increasing width linear and non-linear models get closer to each other) is well established in the NTK setting for supervised models. However such results are stated in terms of a squared loss while the above findings show that the same behaviour for the SSL setting and under orthogonality constraints.

---

[2]This last assumption can also be weakened to say that $\phi''$ is continuous at 0. See the proof of the theorem for details.

## 4 Learning Dynamics of Deep Linear SSL Models

Having showed that the non-linear dynamics are close to the linear ones we now derive the learning dynamics and discuss the evolution of the differential equation. Furthermore we numerically evaluate the theoretical results and show that the dynamics coincide with learning the general loss function under gradient decent. Importantly we now consider *deep* neural networks of depth $l$ such that we now analyze the following trace minimization problem.

**Definition 1** (**General Loss Function**). *Consider the following loss function for a $l$ layer deep linear network*

$$\min_{W_i, \ i \in [l]} \text{Tr}\left(W_l^\top W_{l-1} \cdots W_1 C W_1^\top \cdots W_{l-1}^\top W_l\right) \quad \text{s.t.} \quad W_i^\top W_i = \mathbb{I}, \ \forall i \in [l]. \tag{7}$$

*where $\mathbb{I}$ is an appropriately sized identity matrix, $W_1 \in \mathbb{R}^{h \times d}$, $W_l \in \mathbb{R}^{h \times z}$ , $W_i \in \mathbb{R}^{h \times h} \ \forall i \in 2, \ldots l - 1$ are the trainable weight matrices. $C \in \mathbb{R}^{d \times d}$ is a symmetric, data dependent matrix.*

With the general optimization problem set up we can analyze (7) by deriving the dynamics under orthogonality constraints on the weights, which constitutes gradient descent on the Grassmannian manifold. While orthogonality constraints are easy to initialize the main mathematical complexity arises from ensuring that the constraint is preserved over time. Following Lai et al. (2020), we do so by ensuring that the gradients lie in the tangent bundle of orthogonal matrices.

### 4.1 Theoretical Analysis

In the following we present the dynamics in Theorem 2, followed by the analysis of the evolution of the dynamics in Theorem 3.

**Theorem 2** (**Learning Dynamics in the Deep Linear Setting**). *Assume the general linear trace minimization problem stated in (7) and assume that $C \in \mathbb{R}^{d \times d}$ is a symmetric, data dependent matrices, such that $C = V \Lambda V^\top$ with $V := [v_1, \ldots, v_d]$. Then with $q := \left[u^{\mathbb{I}}(v_1), \cdots, u^{\mathbb{I}}(v_d)\right]^\top$, where $u$ represents the machine function i.e. $u^{\mathbb{I}}(x) = W_l^\top W_{l-1} \cdots W_1 x$, the learning dynamics of $q$, the machine outputs are given by*

$$\mathring{q} = -2\left[2\Lambda q - \Lambda q q^\top q - q q^\top \Lambda q\right]. \tag{8}$$

Similar differential equations to (8) have been analysed in Yan et al. (1994) and Fukumizu (1998). The typical way to find stable solutions to such equations involve converting it to a differential equation on $q q^\top$. This gives us a matrix riccati type equation. For brevity's sake we write below a complete solution when $z = 1$.

**Evolution of the differential equation.** While the above differential equation doesn't seem to have a simple closed form, a few critical observations can still be made about it - particularly about what this differential equation converges to. As observed in Figure 3 (right), independent of initialisation we converge to either of two points. In the following we formalise this observation.

**Theorem 3** (**Evolution of learning dynamics in** (8) **for** $z = 1$). *Let $z = 1$ then our update rule simplifies to*

$$\frac{\mathring{q}}{2} = -(1 - q^\top q)\Lambda q - (\mathbb{I} - q q^\top)\Lambda q. \tag{9}$$

*We can distinguish two cases:*

- *Assume all the eigenvalues of $\Lambda$ are strictly positive then $q$ converges to $0$ with probability $1$.*

- *Assume there is atleast one negative eigenvalue of $\Lambda$, then $q$ becomes the smallest eigenvector, $v_1$.*

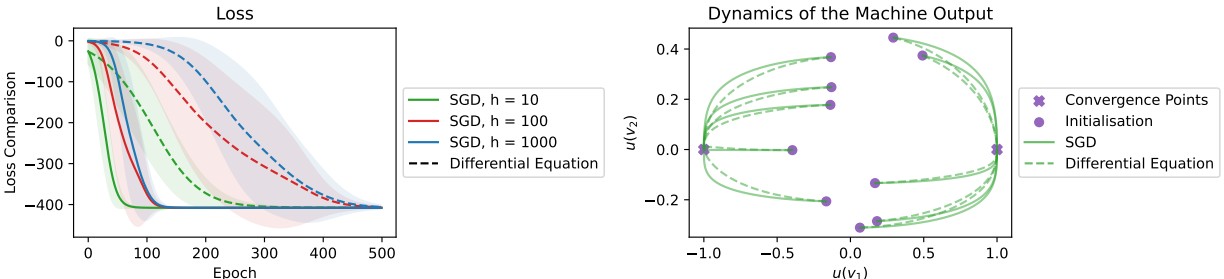

Figure 3: Comparison of gradient decent optimization and differential equation. **(left)** comparison of the loss function **(right)** comparison of the outputs.

The requirement of negative eigenvalues of $C$ for a non-trivial convergence might be surprising however we can observe this when considering $C$ in expectation. Let us assume $C$ is constructed by (5) and note that $\mathbb{E}[\widetilde{C}] = \mathbb{E}\left[\sum_i^n x_i \left(x_i^- - x_i^+\right)^\top\right]$. While this already gives a heuristic of what is going on, for some more precise mathematical calculations, we can specialise to the situation where $x^-$ is given by an independent sample and $x^+$ is given by adding a noise value $\epsilon$ sampled from $N(0, \sigma\mathbb{I})$, i.e. $x^+ = x + \epsilon$. Then

$$\mathbb{E}[\widetilde{C}] = \sum_{i=1}^n \mathbb{E}[x_i]\mathbb{E}[x_i^{-\top}] - \mathbb{E}[x_i x_i^{+\top}] = -n\mathbb{E}[xx^\top].$$

Thus $\mathbb{E}[C]$ is in fact negative definite. Extending Theorem 3 to a general $k > 1$ we conjecture the following to hold.

**Conjecture 2** (**Evolution of learning dynamics in** (8) **for** $z > 1$). *If $C$ has atleast $k$ strictly negative eigenvalues and $v_1, ..., v_k$ are the eigenvectors corresponding to the most negative eigenvalues then the space spanned by $v_1, ..., v_k$ and that by $q_1, ..., q_k$ are the same in limit where $q_i$ denotes the $i$'th column of $q$.*

We note that to prove the conjecture it is enough to show in limit $v_1, ..., v_k$ are each contained in the space spanned by $q_1, ..., q_k$. As the norm of $v_i$ is 1, it is enough in turn to show that the norm of the projection of $v_i$ onto the space of $q_1, ..., q_k$ converges to 1. We experimentally verify and illustrate this in Figure 4.

**New Datapoint**. While the above dynamics provide the setting during training we can furthermore investigate what happens if we input a new datapoint or a testpoint to the machine. Because $u$ is a linear function and because $v_1, ..., v_d$ is a basis this is quite trivial. So if $\hat{x}$ is a new point, let $\alpha = (\alpha_1, ..., \alpha_d)^\top$ be the co-ordinates of $\hat{x}$, i.e. $\hat{x} = \sum_i^d \alpha_i v_i$ or $\alpha = V^\top \hat{x}$. Then

$$u_t(\hat{x}) = u_t\left(\sum_i^d \alpha_i v_i\right) = \sum_i^d \alpha_i u_t(v_i) = q_t^\top \alpha = q_t^\top V^\top \hat{x}.$$

## 4.2 Numerical Evaluation

We can now further illustrate the above derived theoretical results empirically. Let us first consider the setting of 2-layer networks.

**Leaning dynamics (Theorem 2) and new Datapoint.** We can now illustrate that the derived dynamics in (8) do indeed behave similar to learning (7) using gradient decent updates. To analyze the learning dynamics we consider the gradient decent update of (7):

$$W_{1,2}^{(t+1)} = W_{1,2}^{(t)} + \eta \nabla \mathcal{L}_{W_2^{(t)}, W_1^{(t)}} \tag{10}$$

where $W_1^{(t)}, W_2^{(t)}$ are the weights at time step $t$ and $\eta$ is the learning rate as a reference. Practically the constraints in (7) are enforced by projecting the weights back onto $W_2^\top W_2 = \mathbb{I}_z$ and $W_1^\top W_1 = \mathbb{I}_d$ after each

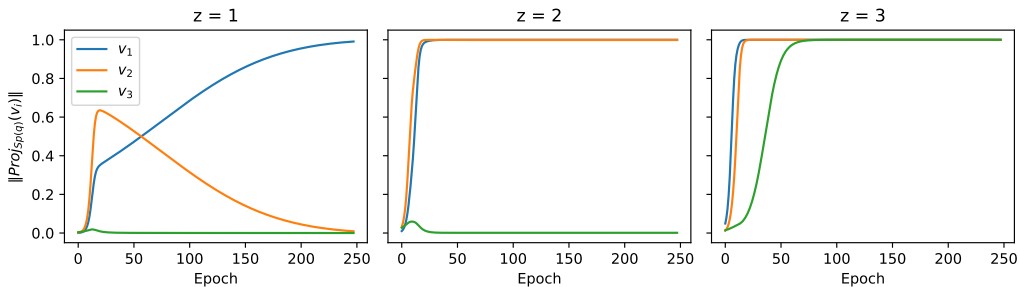

Figure 4: Consider the dataset presented in Figure 1a and embedding into $z = \{1, 2, 3\}$. Plotted is the norm of the projection of $v_1, \ldots v_3$ onto the space spanned by the columns of $q$.

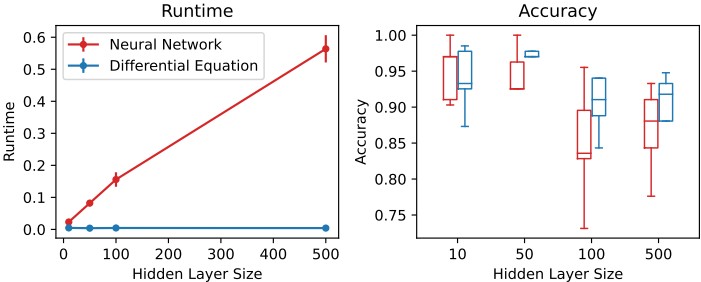

Figure 5: **(left)** Run-time comparison between running differential equation and SGD iteration for different hidden layer width. **(right)** Downstream task: Accuracy comparison for SVM on embedding obtained by SGD optimization and running the differential equation.

gradient step. Secondly we consider a discretized version of (8)

$$q_{t+1} = q_t - \eta \left[ 2\Lambda q_t - \Lambda q_t q_t^\top q_t - q_t q_t^\top \Lambda q_t \right].\tag{11}$$

where $q_t$ is the machine outputs at time step $t$.

We now illustrate the comparison through in Figure 3 where we consider different width of the network ($h \in \{10, 100, 1000\}$) and $\eta = 0.01$. We can firstly observe on the left, that the loss function of the trained network and the dynamics and observe while the decay is slightly slower in the dynamics setting both converge to the same final loss value. Secondly we can compare the function outputs during training in Figure 3 (right): We initialize the NN randomly and use this initial machine output as $q_0$. We observe that during the evolution using (10) & (11) for a given initialization the are stay close to each other and converge to the same final outputs.

**Numerical Evaluation of Theorem 3.** We can again illustrate that the behaviour stated in Theorem 3 can indeed be observed empirically. This is shown in Figure 3 (right), a setting where $C$ has negative eigenvalues. We observe that eventually the machine outputs converge to the smallest eigenvector.

**Runtime and downstream task.** Before going into the illustration of the dynamics we furthermore note that an update step using (11) is significantly faster then a SGD step using (10). For this illustration we now consider two classes with 200 datapoints each from the MNIST dataset Deng (2012). This is illustrated in Figure 5 (left) where we compare the runtime over different layer width (of which (11) is independent of). Expectantly (10) scales linearly with $h$ and overall (11) has a shorter

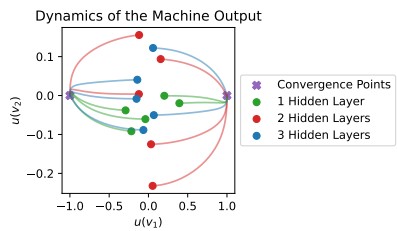

Figure 6: Numerical illustration of the convergence of deep linear models.

runtime per timestep. While throughout the paper we focus on the obtained embeddings we can furthermore consider the performance of downstream tasks on top of the embeddings. We illustrate this in the setting above where we apply a linear SVM on top of the embeddings. The results are shown in Figure 5 (right) where we observe that overall the performance of the downstream task for both the SGD optimization and the differential equation coincide.

**Deep Linear Networks.** Numerically we illustrate that the learning dynamics as derived in Theorem 2 for several hidden layer sizes. In Figure 6 we observe that for one, two and three hidden layers the dynamics behave very similarly and all converge to the same final embedding, the one derived in Theorem 3.

## 5 Non-Contrastive Loss Functions

In the previous sections analyzed contrastive loss functions however as we show in this section the derived results extend beyond this setting and are also applicable to con-contrastive losses. Non-contrastive losses emerged from the observation that negative samples (or pairs) in contrastive SSL are not necessary in practice, and it suffices to maximise only alignment between positive pairs Chen & He (2021); Chen et al. (2020); Grill et al. (2020). Considering a simplified version of the setup in Chen et al. (2020) one learns a representation by minimising the loss [3]

$$\min_{\Theta} \sum_{i=1}^{n} -u(x_i)^{\top} u(x_i^+). \tag{12}$$

The embedding $u = [u_1(\cdot, \Theta) \dots u_z(\cdot, \Theta)]^{\top} : \mathbb{R}^d \to \mathbb{R}^z$, parameterised by $\Theta$, typically comprises of a base encoder network and a projection head in practice (Chen et al., 2020). We now shortly outline how the previously presented results apply to this setting as well.

**Dimension collapse.** Analogous to the contrastive setting earlier we can now discuss dimension collapse for the non-contrastive setting. Assuming that the network function $u : \mathbb{R}^d \to \mathbb{R}^z$ is parameterised by $\Theta$, the gradient of the loss $\mathcal{L}(\Theta) = \sum_{i=1}^{n} -u(x_i)^{\top} u(x_i^+)$ is $\nabla_{\Theta} \mathcal{L}(\Theta) = -\sum_{i=1}^{n} \sum_{j=1}^{z} u_j(x_i) \cdot \nabla_{\Theta} u_j(x_i^+) + u_j(x_i^+) \cdot \nabla_{\Theta} u_j(x_i)$. Hence, under gradient descent $\mathring{\Theta} = -\nabla_{\Theta} \mathcal{L}$, the evolution of each component of $u(x)$, given by $\mathring{u}_l(x) = \left\langle \nabla_{\Theta} u_l(x), \mathring{\Theta} \right\rangle$ is $\mathring{u}_l(x) = \sum_{i=1}^{n} \sum_{j=1}^{z} \left\langle \nabla_{\Theta} u_l(x), \nabla_{\Theta} u_j(x_i) \right\rangle u_j(x_i^+) + \left\langle \nabla_{\Theta} u_l(x), \nabla_{\Theta} u_j(x_i^+) \right\rangle u_j(x_i)$. For linear model, $u(x) = \Theta x$, the dynamics of $u(x)$ is given by $\mathring{u}_l(x) = \sum_{i=1}^{n} (x^{\top} x_i) u_l(x_i^+) + (x^{\top} x_i^+) u_l(x_i)$. From there Proposition 1 extends to the non-contrastive setting. Under the conditions of Lemma 1, every component of the network output $u : \mathbb{R}^d \to \mathbb{R}^z$ has identical dynamics. As a consequence, the output collapses to one dimension at convergence.

**Learning dynamics of linear non-contrastive models.** In addition we can also frame[4] the previously considered non-contrastive model in (12) in the simple linear setting by considering the general loss function with $\widetilde{C} = \sum_{i}^{n} x_i \left( x_i^+ \right)^{\top}$. Therefore the results from Theorem 2 & 3 extend to models under (12) as well.

**Limitations and trivial solutions.** For arbitrary functions this loss can have a trivial minimizer that is constant function. However in the linear setting we mainly consider in this paper the only constant function is the zero function which would not be optimal.

## 6 Conclusion

The study of learning dynamics of (infinite-width) neural networks has led to important results for the supervised setting such as understanding the loss landscape and convergence, early stopping, linearised (kernel) approximations and, mostly importantly, generalisation and inductive biases. The analysis of similar quantities is of interest in the SSL setting as well, however, there is little understanding of SSL dynamics.

---

[3]We simplify Chen et al. (2020) by replacing the cosine similarity with the standard dot product and also by replacing an additional positive sample $x_i^{++}$ by anchor $x_i$ for convenience.

[4]It has previously been observed that one can unify contrastive and non-contrastive losses in a more general framework (Garrido et al., 2022).

Our initial steps towards analysing SSL dynamics encounters a hurdle: standard SSL training has drastic dimension collapse (Proposition 1), unless there are suitable constraints. We consider a general formulation of linear SSL under orthogonality constraints (7), and derive its learning dynamics (Theorem 2). We also show that the derived dynamics can approximate the SSL dynamics using wide neural networks (Theorem 1) under some conditions on activation $\phi$. We not only provide a framework for analysis of SSL dynamics, but also shows how the analysis can critically differ from the supervised setting. As we numerically demonstrate, our derived dynamics can be used an efficient computational tool to approximate SSL models. In particular, the equivalence in Proposition 3 ensures that the orthogonality constraints can be equivalently imposed using a scaled loss, which is easy to implement in practice. We conclude with a limitation and open problem. Our analysis relies on a linear approximation of wide networks, but more precise characterisation in terms of kernel approximation (Jacot et al., 2018; Liu et al., 2020) may be possible, which can better explain the dynamics of deep SSL models. However, integrating orthogonality or operator norm constraints in the NTK regime remains an open question.

### Acknowledgements

This work has been supported by the German Research Foundation (DFG) through the Priority Program SPP-2298 (project GH-257/2-1) and DFG-ANR PRCI "ASCAI" (GH 257/3-1).

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

# A  Appendix

In the appendix we provide the following additional proofs

- Proof of Lemma1

- Proof of Proposition 2

- Proof of Proposition 3

- Proof of Theorem 1

- Proof of Theorem 2

- Proof of Theorem 3

## A.1  Proof of Lemma1

*Proof.* Let the collumns of $W_2$ be denoted by $w_1, w_2, ..., w_z$. Then we note that each component of $u$, $u_j$ is given by $u_j(x) = w_j^\top \phi(W_1 x)$. Thus if $l \neq j$, $u_j(x)$ has no dependence with $w_l$ i.e. $\nabla_{w_l} u_j(x) = 0$. Thus we get that when $l \neq j$,

$$\langle \nabla_\Theta u_l(x), \nabla_\Theta u_j(x') \rangle = \langle \nabla_{W_1} u_l(x), \nabla_{W_1} u_j(x') \rangle.$$

We can now use Liu et al. (2020) (for instance its Lemma 3.1) which basically concludes that no training happens at the penultimate or prior layers. In limit all positive gradients arise only from the final layer. As such

$$\langle \nabla_{W_1} u_l(x), \nabla_{W_1} u_j(x') \rangle = 0.$$

By the same token, for $l = j$,

$$\langle \nabla_\Theta u_l(x), \nabla_\Theta u_j(x') \rangle = \langle \nabla_{W_1} u_l(x), \nabla_{W_1} u_j(x') \rangle + \langle \nabla_{w_j} u_j(x), \nabla_{w_j} u_j(x') \rangle$$
$$= \langle \phi(W_1 x), \phi(W_1 x') \rangle.$$

Finally again using the fact that $W_1$ does not change in training and that $W_1$ is initialized from a normalized gaussian , when $\phi$ is the identity map, it is well known that the above converges to $x^\top x'$ (as there $\langle \phi(W_1 x), \phi(W_1 x') \rangle = x^\top (W_1^\top W_1) x \to x^\top x'$) and otherwise to a deterministic kernel $k$ (see e.g. (Liu et al., 2020), (Arora et al., 2019b)). $\square$

## A.2  Proof of Proposition 2

*Proof.* For simplicity of the proof we begin by reformulating the loss function in both contrastive and noncontrastive setting to a more general form. In particular it is trivial to check that we can generalize by writing
$$\mathcal{L} = \text{Tr}\left(W_2^\top f(X, W_1) W_2\right),$$

where $X$ denotes the collection of all the relevant data (i.e. $\forall\ 1 \leq i \leq n\ x_i$, as well as $x_i^+$ and $x^-$ where applicable), and $f(X, W_1) = \sum_{i=1}^n \phi(W_1 x_i)\left(\phi(W_1 x_i^-) - \phi(W_1 x_i^+)\right)^\top$ in the contrastive setting (equation 1) while $f(X, W_1) = -\sum_{i=1}^n \phi(W_1 x_i)\phi(W_1 x_i^+)^\top$ in the non-contrastive setting (equation 12.)

Then decompose
$$W_2 W_2^\top = \sum_{i=1}^k \sigma_i^2 v_i v_i^\top.$$

Note then that $\|W_2\|_F^2 = \mathrm{Tr}\left(W_2 W_2^\top\right) = \sum_{i=1}^k \sigma_i^2$. Thus the optimization target,

$$\mathcal{L}(W_1, W_2) = \mathrm{Tr}\left(W_2^\top f(X, W_1) W_2\right) = \mathrm{Tr}\left(f(X, W_1) W_2 W_2^\top\right) = \mathrm{Tr}\left(f(X, W_1) \sum_{i=1}^k \sigma_i^2 v_i v_i^\top\right)$$

$$= \sum_{i=1}^k \sigma_i^2 v_i^\top f(X, W_1) v_i \geq \min_{i=1 \ to \ k} \{v_i^\top f(X, W_1) v_i\} \sum_{i=1}^k \sigma_i^2 = \|W_2\|_F^2 \min_{i=1 \ to \ k} \{v_i^\top f(X, W_1) v_i\}.$$

Thus when the Frobenius norm is restricted (i.e. bounded between 0 and $c$), if $f(X, W_1)$ has atleast one negative eigenvalue the loss is minimized when $v_1$ is the eigenvector corresponding to the most negative eigenvalue of $f(X, W_1)$ with $\sigma_1 = \|W_2\|_F$, with no other non-zero singular value. On the other hand if $f(X, W_1)$ has no negative eigenvalue then the loss is minimized when $W_2 = 0$. $\qquad\square$

### A.3 Proof of Proposition 3

*Proof.* We begin by quickly observing that (1) $\iff$ (2). This is simply done by defining $\hat{W}_i = \frac{W_i}{\|W_i\|_2}$ for $i = 1, 2$. Then we have

$$\underset{W_1, W_2}{\mathrm{argmin}} \frac{\mathrm{Tr}\left(W_2^\top W_1 C W_1^\top W_2\right)}{\|W_1\|_2^2 \|W_2\|_2^2} = \underset{\hat{W}_1, \hat{W}_2 : \|\hat{W}_1\|_2 = \|\hat{W}_1\|_2 = 1}{\mathrm{argmin}} \mathrm{Tr}\left(\hat{W}_2^\top \hat{W}_1 C \hat{W}_1^\top \hat{W}_2\right)$$

Using the fact that atleast one eigenvalue of $C$ is strictly negative (this rules out the case that the optimal is achieved when $W_i = 0$ as that would have prevented division by norm) then we can quickly get that

$$\underset{\hat{W}_1, \hat{W}_2 : \|\hat{W}_1\|_2 = \|\hat{W}_1\|_2 = 1}{\mathrm{argmin}} \mathrm{Tr}\left(\hat{W}_2^\top \hat{W}_1 C \hat{W}_1^\top \hat{W}_2\right) = \underset{\hat{W}_1, \hat{W}_2 : \|\hat{W}_1\|_2 \leq 1; \|\hat{W}_1\|_2 \leq 1}{\mathrm{argmin}} \mathrm{Tr}\left(\hat{W}_2^\top \hat{W}_1 C \hat{W}_1^\top \hat{W}_2\right).$$

For (2) $\iff$ (3), we begin by observing that by submultiplicativity of norm, any $W_1, W_2$ such that $\|W_1\|_2 \leq 1$ and $\|W_2\|_2 \leq 1$ automatically falls is the optimization space given by $\|W_1^\top W_2\| \leq 1$ thus giving one direction of the optimization equivalence for free. For the other side we note that given any $W_1, W_2$ such that $\|W_1^\top W_2\|_2^2 = \|W_1^\top W_2 W_2^\top W_1\|_2 \leq 1$, we can construct $\hat{W}_1, \hat{W}_2$ such that $\left\|\hat{W}_i\right\| \leq 1$ and $W_1^\top W_2 W_2^\top W_1 = \hat{W}_1^\top \hat{W}_2 \hat{W}_2^\top \hat{W}_1$. This follows from considering the singular values decomposition of $W_1^\top W_2$, getting $W_1^\top W_2 = U^\top \Sigma V$. As the norm of the product is smaller than 1, all the entries of the singular value matrix $\Sigma$ are less than 1. Thus depending upon which among $d$ or $z$ is larger we consider either the matrices $\Sigma U$ and $V$ or the matrices $U$ and $\Sigma V$ to be our candidate $\hat{W}_1$ and $\hat{W}_2$ respectively. To complete we will simply have to add zero rows to our choice i.e. say $U$ and $\Sigma V$ to match the dimensions (i.e. to get a $n \times d$ matrix from a $z \times d$ one).

Finally for (3) $\iff$ (4) we begin by defining $W = W_1^\top W_2$. Then the optimization problem in (3) becomes,

$$\min_{W : \|W\|_2 \leq 1} \mathrm{Tr}\left(W^\top C W\right) = \min_{W : \|W\|_2 \leq 1} \mathrm{Tr}\left(C W W^\top\right).$$

We then prove that we are done if we can prove the claim at optimal of (3) (i.e. the above optimization problem) all the eigenvalues of $W W^\top$ are 1 or 0. Given this claim the singular value decomposition of $W$ becomes only $W = U^\top V$, where if $k = rank(W)$, $U$ is a $k \times d$ matrix and $V$ a $k \times z$ matrix. Additionally by property of SVD, the collumns of $U$ and $V$ are orthonormal. Finally as

$$k = rank(W) \leq \min\{rank(W_1), rank(W_2)\} \leq \min\{d, z\} \leq n,$$

we can add a bunch of zero rows to $U$ and $V$ to get our $n \times d$ and $n \times z$ matrices which will be our corresponding $W_1$ and $W_2$.

It remains to prove that $\mathrm{Tr}\left(C W W^\top\right)$ is minimized when all the eigenvalues of $W W^\top$ are 1 or 0. To do this simply decompose

$$W W^\top = \sum_{i=1}^k \sigma_i^2 v_i v_i^\top,$$

where $v_i$ is the set of orthonormal eigenvectors of $WW^\top$ corresponding to non-zero eigenvalues of $WW^\top$ (or alternatively non-zero singular values of $W$) Then

$$
\begin{aligned}
\operatorname{Tr}\left(CWW^\top\right) &= \operatorname{Tr}\left(C\sum_{i=1}^{k}\sigma_i^2 v_i v_i^\top\right) \\
&= \sum_{i=1}^{k}\sigma_i^2 \operatorname{Tr}\left(Cv_i v_i^\top\right) \\
&= \sum_{i=1}^{k}\sigma_i^2 v_i^\top C v_i.
\end{aligned}
$$

Thus if $C$ has $l$ many strictly negative eigenvalues $\lambda_1 \leq \cdots \leq \lambda_l$ with corresponding eigenvectors $c_1,\ldots,c_l$ and $\sigma_i^2$ is positive the above quantity is minimized by choosing as many of these as possible i.e. $v_1 = c_1,\ldots,v_{\min\{d,z,l\}} = c_{\min\{d,z,l\}}$ and setting the corresponding $\sigma_i$ to be 1 while every setting all other eigenvalues to 0.

We then also note by consequence of the above proof that we avoid dimension collapse when possible i.e. when $C$ has multiple strictly negative eigenvalues (which is what one should expect if the data is not one dimensional as $\mathbb{E}[C] = -\mathbb{E}[xx^\top]$) $\qquad\square$

### A.4 Proof of Theorem 1

*Proof.* Let us start by defining some properties for the non-linearity: Assume the non-linear function $\phi$ is continuously twice differentiable near 0 and has no bias i.e. $\phi(0) = 0$. Then via scaling we can assume WLOG that $\phi'(0) = 1$. As $|\phi''(x)| \leq c$, we get that [5]

$$
|\phi(x) - x| \leq \frac{cx^2}{2}. \tag{13}
$$

Recall that the mapping of the first weight matrix is given by $W_1 : \mathbb{R}^d \to \mathbb{R}^h, \quad h \gg d$ under the constraint that $W_1^\top W = \mathbb{I}$. Under uniformly random initialization by Lemma 2 (see proof below) then with probability asymptotically going to 1 we have that

$$
\max (W_1)_{i,j}^2 \leq C\frac{\log^2 h}{h}
$$

Thus the norm of each row of $W_1$ we get with a.w.h.p. :

$$
\|\operatorname{row}_i (W_1)\|^2 = \sum_{j=1}^{d}(W_1)_{i,j}^2 \leq C\frac{d\log^2 h}{h}
$$

From there we can now write the value of each node in the layer using Cauchy-Schwarz inequality as

$$
|\operatorname{row}_i(W_1) \cdot x|^2 \leq \|\operatorname{row}_i (W_1)\|^2 \|x\|^2 \leq C\|x\|^2\frac{d\log^2 h}{h}. \tag{14}
$$

We now apply the non-linearity to this quantity and denote the output of the first layer after the non-linearity as

$$
v_i = \phi\left(\operatorname{row}_i (W_1) \cdot x\right)
$$

Define the vector $\epsilon \in \mathbb{R}^h$, where

$$
\epsilon_j = v_i - \operatorname{row}_i (W_1) \cdot x
$$

---

[5]We can actually also use the weaker assumption that $\phi''(0)$ is continuous at 0. Thus there is some bounded (compact) set $A$ containing 0 and a constant $c$ such that $\forall x \in A$, $|\phi(x) - x| \leq \frac{cx^2}{2}$

Then we have for $h$ large enough[6]:

$$\|\epsilon\|^2 = \sum_{i=1}^{h} \epsilon_i^2$$

$$= \sum_{i=1}^{h} (v_i - \mathrm{row}_i(W_1) \cdot x)^2$$

$$\leq \sum_{i=1}^{h} \frac{c^2}{4} (\mathrm{row}_i(W_1) \cdot x)^4 \qquad\qquad \textit{by equation 13}$$

$$\leq \sum_{i=1}^{h} \frac{c^2}{4} \left( C \|x\|^2 \frac{d \log^2 h}{h} \right)^2 \qquad\qquad \textit{by equation 14}$$

$$= K^2 c^2 \|x\|^4 \frac{h d^2 \log^4 h}{h^2} = K^2 c^2 \|x\|^4 \frac{d^2 \log^4 h}{h},$$

where $K$ is the universal constant $\frac{C}{2}$. Combining this with the second layer we get the difference of the outputs of the two networks as

$$\left\| u_{(0)}^\phi - u_{(0)}^{\mathbb{I}} \right\| = \left\| W_2^\top v - W_2^\top W_1 x \right\|$$

$$= \left\| W_2^\top (v - W_1 x) \right\|$$

$$\leq \|W_2\| \|\epsilon\| = \|\epsilon\| \qquad\qquad \textit{as } \|W_2\| = 1$$

$$\leq K c \|x\|^2 d \sqrt{\frac{\log^4 h}{h}}$$

$$\to 0.$$

$\square$

**Lemma 2.** *Given any $d \leq p$, Let $Q$ be a uniformly random $h \times d$ semi-orthonormal matrix. I.e. $Q$ is the first $d$ columns of an uniformly random $h \times h$ orthonormal matrix. Then there are constants $L$ and a sequence $\epsilon_p$ converging to $0$ as $h$ goes to infinity such that ,*

$$P \left( \max |Q_{i,j}| \geq \frac{L \log h}{\sqrt{h}} \right) \leq \epsilon_n$$

*Proof.* We note that it is enough to prove the claim when $d = h$, i.e. $Q$ is uniformly random $h \times h$ orthonormal matrix. Then as our distribution is uniform, the density at any particular $Q$ is same as the density at any $UQ$ where $U$ is some other fixed orthogonal matrix. Thus if $q_1$ is the first column of $Q$, the marginal distribution of $q_1$ has the property that its density at any $q_1$ is same as that of $Uq_1$ for any orthogonal matrix $U$. In other words the marginal distribution for any column of $Q$ is simply that of the uniform unit sphere.

Consider then the following random variable which has the same distribution as that of a fixed column of $Q$ i.e. uniform unit $h$-sphere. Let $X = (X_1, ..., X_h)$ be iid random variables from $\mathcal{N}(0,1)$. Then we know that $X \sim \mathcal{N}(0, \mathbb{I}_h)$. From the rotational symmetry property of standard gaussian then we have that $\frac{X}{\|X\|}$ is distributed as an uniform sample from the unit sphere in $h$ dimensions. By union bound then, we have

$$P \left( \max_{1 \leq i \leq h} |X_i| \geq t \log h \right) \leq \frac{1}{\sqrt{2\pi}} h e^{-\frac{t^2 \log^2 h}{2}}$$

$$\implies P \left( \max_{1 \leq i \leq h} |X_i| \leq t \log h \right) \geq 1 - \frac{1}{\sqrt{2\pi}} h e^{-\frac{t^2 \log^2 h}{2}}.$$

---

[6]Note that for the weaker assumption we can still use equation 13. This is because by equation 14,w.h.p. $\mathrm{row}_i(W_1) \cdot x$ goes to 0 and thus $\mathrm{row}_i(W_1) \cdot x \in A$ in limit

As each $X_i$ is iid normal, $X_i^2$ is iid Chi-square with $\mathbb{E}[X_i^2] = 1$, thus by Chernoff there exists constants $C', c'$ such that

$$P\left(\frac{\sum_{i=1}^h X_i^2}{h} \geq 1 - s\right) \geq 1 - C' e^{-c' h s^2}.$$

Since $\max_{1 \leq i \leq h} |X_i| \leq t \log h$ and $\frac{\sum_{i=1}^h X_i^2}{h} \leq (1+s)$ implies that $\max_{1 \leq i \leq h} \frac{|X_i|}{\|X\|} \leq \frac{t \log h}{\sqrt{h(1-s)}}$, we get that

$$P\left(\max_{1 \leq i \leq h} \frac{|X_i|}{\|X\|} \leq \frac{t \log h}{\sqrt{h(1-s)}}\right) \geq 1 - \frac{1}{\sqrt{2\pi}} h e^{-\frac{t^2 \log^2 h}{2}} - C' e^{-c' h s^2}$$

$$\implies P\left(\max_{1 \leq i \leq h} \frac{|X_i|}{\|X\|} \geq \frac{t \log h}{\sqrt{h(1-s)}}\right) \leq \frac{1}{\sqrt{2\pi}} h e^{-\frac{t^2 \log^2 h}{2}} + C' e^{-c' h s^2}$$

From the argument before that any $j$'th column of $Q$ is distributed as $X$. Using the above and another union bound then get us

$$P\left(\max_{1 \leq i \leq h} \max_{1 \leq i \leq h} |Q_{i,j}| \geq \frac{t \log h}{\sqrt{h(1-s)}}\right) \leq \frac{1}{\sqrt{2\pi}} h e^{-\frac{t^2 \log^2 h}{2}} + C' e^{-c' h s^2}$$

$$\implies P\left(\max_{1 \leq j \leq h} \max_{1 \leq i \leq h} |Q_{i,j}| \geq \frac{t \log h}{\sqrt{h(1-s)}}\right) \leq \frac{1}{\sqrt{2\pi}} h^2 e^{-\frac{t^2 \log^2 h}{2}} + C' h e^{-c' h s^2}$$

We note that for any constants $t, c'$ that as $h$ goes to infinity, both $h^2 e^{-\frac{t^2 \log^2 h}{2}}$ and $h e^{-c' h s^2}$ goes to zero. The proof is then finished by choosing some appropriate constants $s, t \geq 0$. □

## A.5  Proof of Theorem 2

*Proof.* To simplify notation we are dropping the superscript $\mathbb{I}$ from $u_{(t)}^{\mathbb{I}}$. The $u$ in the following proof is already presumed to be linear. For the same reason we are also dropping the symbol of time, $t$, from $u, W_3, W_2, W_1$ even though all of them are indeed time dependent. Finally for any time dependent function $f$, we denote $\frac{\partial f}{\partial t}$ by $\mathring{f}$.

Using this and recalling that the loss in Eq. 7. Consider the following loss function for a $l$ layer deep linear network

$$\min_{W_i, \ i \in [l]} \text{Tr}\left(W_l^\top W_{l-1} \cdots W_1 C W_1^\top \cdots W_{l-1}^\top W_l\right) \quad \text{s.t.} \quad W_i^\top W_i = \mathbb{I}, \ \forall i \in [l].$$

where $\mathbb{I}$ is an appropriately sized identity matrix, $W_1 \in \mathbb{R}^{h \times d}$, $W_l \in \mathbb{R}^{h \times z}$, $W_i \in \mathbb{R}^{h \times h} \ \forall i \in 2, \ldots l - 1$ are the trainable weight matrices. $C \in \mathbb{R}^{d \times d}$ is a symmetric, data dependent matrix.

To simplify the proof we now consider a three hidden layer network and we will observe that the obtained results extends to arbitrarily deep networks. Let the embedding function therefore be

$$u(x) := W_3^T W_2 W_1 x \quad \text{s.t.} \quad W_1^\top W_1 = \mathbb{I}, W_3^\top W_3 = \mathbb{I} \text{ and } W_2^\top W_2 = \mathbb{I}.$$

From (Edelman et al., 1998), we get that the derivative of a function $\gamma$ restricted to a grassmanian is derived by left-multiplying $1 - \gamma\gamma^\top$ to the "free" or unrestricted derivative of $\gamma$. We therefore can write $\mathring{W}_1, \mathring{W}_2$ and $\mathring{W}_3$ as

$$\mathring{W}_3(t) = -\left(\mathbb{I} - W_3 W_3^\top\right) \nabla_{W_3} \mathcal{L} = -2\left(\mathbb{I} - W_3 W_3^\top\right)\left(W_2 W_1 C W_1^\top W_2^\top W_3\right),$$
$$\mathring{W}_2(t) = -\left(\mathbb{I} - W_2 W_2^\top\right) \nabla_{W_2} \mathcal{L} = -2\left(\mathbb{I} - W_2 W_2^\top\right)\left(W_3 W_3^\top W_2 W_1 C W_1^\top\right),$$
$$\mathring{W}_1(t) = -\left(\mathbb{I} - W_1 W_1^\top\right) \nabla_{W_1} \mathcal{L} = -2\left(\mathbb{I} - W_1 W_1^\top\right)\left(W_2^\top W_3 W_3^\top W_2 W_1 C\right).$$

Thus we obtain

$$
\begin{aligned}
\frac{\partial\, u_{(t)}(x)}{\partial\, t} =& \mathring{W}_2(t)^\top W_2 W_1(t)x + W_3(t)^\top \mathring{W}_2 W_1(t)x + W_3(t)^\top W_2 \mathring{W}_1(t)x \\
=& \left(\left(\mathbb{I} - W_3 W_3^\top\right)\left(-W_2 W_1 C W_1^\top W_2^\top W_3\right)\right)^\top W_2 W_1(t)x \\
& + W_3(t)^\top \left(\mathbb{I} - W_2 W_2^\top\right)\left(-W_3 W_3^\top W_2 W_1 C W_1^\top\right) W_1(t)x \\
& + W_3(t)^\top W_2 \left(\mathbb{I} - W_1 W_1^\top\right)\left(-W_2^\top W_3 W_3^\top W_2 W_1 C\right)x \\
=& -2\left(2 W_3^\top W_2 W_1 C x\right) \\
& + 2\left(W_3^\top W_2 W_1 C W_1^\top W_2^\top W_3 W_3^\top W_2 W_1 x \ + \ W_3^\top W_2 W_1 W_1^\top W_2^\top W_3 \ W_3^\top W_2 W_1 C x\right) \\
=& -2\left(2 W_3^\top W_2 W_1 C x - W_3^\top W_2 W_1 C W_1^\top W_2^\top W_3 W_3^\top W_2 W_1 x\right. \\
& \left. - \sum_i^d W_3^\top W_2 W_1 v_i v_i^\top W_1^\top W_2^\top W_3 \ W_3^\top W_2 W_1 C x\right),
\end{aligned}
$$

where we obtain the second equality by expanding the terms, taking advantage of $W_i^\top W_i = \mathbb{I}$, $i \in [3]$, and $\mathbb{I}_d = \sum_i^d v_i v_i^\top$. Now setting $x$ as $v_j$ and using the fact that they are eigenvectors for $C$ and using $C = \sum_i^d \lambda_i v_i v_i^\top$ gives us:

$$
\mathring{u}(v_j) = -2\left(2\lambda_j u_{(t)}(v_j) - \sum_i^d \lambda_i u_{(t)}(v_i) u_{(t)}(v_i)^\top u_{(t)}(v_j) - \lambda_j \sum_i^d u_{(t)}(v_i) u_{(t)}(v_i)^\top u_{(t)}(v_j)\right)
$$

Let's rewrite this in matrix notation. First define $q := [u(v_1), \dots u(v_d)]^\top$ thus obtaining:

$$
\mathring{q} = -2\left[2\Lambda q - \Lambda q q^\top q - q q^\top \Lambda q\right].
$$

Finally note that the presented argument would apply similarly for any additional hidden layer, through the same formulation as $W_2$, therefore extending the result to deeper networks as well which concludes the proof.

$\square$

## A.6 Proof of Theorem 3

*Proof.* For instance first suppose that all the eigenvalues of $\Lambda$ are strictly positive and thus $q^\top \Lambda q > 0$. Then

$$
\begin{aligned}
\frac{d(q^\top q)}{dt} &= 2q^\top \mathring{q} = 4\left[-(1 - q^\top q)q^\top \Lambda q - q^\top(\mathbb{I} - q q^\top)\Lambda q\right] \\
&= -8(1 - q^\top q)q^\top \Lambda q
\end{aligned}
$$

Observing now that because of orthonormality of our weight matrices, $q^\top q = \|q\|^2 < 1$ (as $\|q\| \neq 1$ at initialization with probability 1) we get that the derivative of $\|q\|^2$ is always negative and thus $q$ converges to 0.

Now suppose on the other hand there is atleast one negative eigenvalue. WLOG let $e_1$ denote the eigenvector with the smallest eigenvalue (which is negative). Then

$$
\begin{aligned}
\frac{d(e_1^\top q)}{dt} &= e_1^\top \mathring{q} = 2\left[-(1 - q^\top q)e_1^\top \Lambda q - e_1^\top(\mathbb{I} - q q^\top)\Lambda q\right] \\
&= 2\left[(1 - q^\top q)(-\lambda_1)e_1^\top q + (q^\top \Lambda q - \lambda_1)e_1^\top q)\right]
\end{aligned}
$$

We now note that $q^\top \Lambda q - \lambda_1 \geq 0$ as $\lambda_1$ is the smallest eigenvalue. Thus as $-\lambda_1$ is positive, the derivative of $e_1^\top q$ is always positive unless $1 - q^\top q = q^\top \Lambda q - \lambda_1 = 0$, which only happens at $q = e_1$. In other words, eventually $q$ becomes the smallest eigenvector $e_1$. $\square$

