# OpenReview forum: "Representation Learning Dynamics of Self-Supervised Models"
_TMLR — Accepted by TMLR_

### Review · Reviewer_w2XZ · 2024-02-16

**Summary Of Contributions:**

This paper studies dimensional collapse in self-supervised methods, where dimensional collapse is defined as having rank-deficient representations.
This paper analyses the contrastive learning and non-contrastive losses and shows that they both suffer from dimension collapse, and proposes an orthogonality constraint that is meant to prevent dimensional collapse.

Theoretical analyses in this paper are under the infinite width limit.
The paper first shows that where the nonlinear dynamics and the linear dynamics are close to each other: consider a linear network $u^\mathbb{I}(x) = W_1^\top W_1x$ and a nonlinear network $u^\phi(x) = W_2^\top \phi(W_1)x$.
- Theorem 1 states that at initialization, the difference in outputs of a linear and a non-linear network is bounded by $O(\|x\|^2 \cdot \frac{d\log^2 h}{\sqrt{h}})$, where $d$ is the feature dimension, and $h$ is the network's hidden dimension.
- There's no formal guarantee for the full training dynamics, but the paper conjectures (Conjecture 1) that the two outputs will remain close throughout training, approaching 0 as $h \rightarrow \infty$.

Given the closeness between the linear and nonlinear case, the training dynamics focuses on the linear case, i.e. $u(x) = W_2^\top W_1x$. The contrastive and non-contrastive objectives are rephrased and unified to a general trace minimization problem with some matrix $C$. The analyses track the evolution of the representations for eigenvectors of $C$  (Theorem 2) and a 1-dimensional special case (Theorem 3).

The paper corroborates the theoretical results with experiments on half-moon and MNIST.

**Audience:**

Yes

**Broader Impact Concerns:**

There's no ethical concerns.

**Claims And Evidence:**

Yes

**Requested Changes:**

One concern I have is that the theoretical results are based on 2-layer linear network, with limited implications to other setups or practice. Please comment more on the implications, or provide results from more complex empirical setups as further evidence.

Please add more thorough discussions to the prior work; please see the comments above for some examples.

I'd also suggest a careful proofread of the manuscript as there are many typos. Here are some examples:
- the section title of Sec 2.2 should be "a trivial solution" or "trivial solutions".
- Thm 1, "a universal constant". Thm 2 contains a typo too.
- Sec 3.2,  "three activation functions" (rather than loss functions)?
- sentence breaks, e.g. in the paragraph below eq (12).
- Prop 2: has $a(x)$ been defined?

**Strengths And Weaknesses:**

Questions:
- I wonder why the paper considers this simplified form of the non-contrastive loss, which has a trivial minimizer that is constant function. The authors cite previous papers to argue that "it suffices to maximise only alignment between positive pairs". However, this misses a crucial point that the success of non-contrastive loss relies on regularizations (e.g. requiring the feature covariance to be close to the identity matrix) or algorithmic tricks (e.g. normalization layers) to preserve the rank of the features.
- Prop 3: I'm not sure why 1&2 can avoid dimensional collapse. For example, if $C$ only has 1 strictly negative eigenvalue, then $W_1,W_2$ only need to be of rank 1. This also means that the optimization problems e.g. 2&4 are not equivalent: the loss functions have the same minimal value, but the set of minimizers are different (e.g. rank-deficient solutions are allowed in loss 2 but not loss 4).
- Conjecture 1: what's the basis for this conjecture? Sec 3.2 provides numerical evaluation, but I don't understand what this adds beyond the intuition from NTK.
- Sec 3.2 / Fig 2: could you clarify the takeaway here please? For example, the results on tanh and ReLU are supportive of a NTK / lazy training point of view (i.e. increasing the layer width makes the linear and non-linear closer to each other), and I don't see how it's specific to SSL.


Relations to prior work should be discussed further, and there are missing citations:
- For theoretical analyses of optimization of contrastive and non-contrastive methods:
  - Wen and Li 21: Toward Understanding the Feature Learning Process of Self-supervised Contrastive Learning
  - Pokle et al. 22: Contrasting the landscape of contrastive and non-contrastive learning
- Barlow Twins, a non-contrastive method whose objective is very similar to the one in Conjecture 1. Notably, the minimizer for the Barlow Twins objective is for the network outputs to have an identity covariance.
- In Section 4, about viewing contrastive and non-contrastive objectives as a "general trace minimization problem": the connection between non-contrastive objectives and trace minimization has been pointed out in prior work, such as Balestriero & LecCun 22. This can also be connected to contrastive methods, e.g. Garrido et al. 22 has observed the duality between contrastive and non-contrastive methods.
  -  Garrido et al. 22: On the duality between contrastive and non-contrastive self-supervised learning.

---

> ### Author Response · Authors · 2024-04-08
>
> We thank the reviewer for the constructive feedback. We address the concerns below.
>
> ### Questions
>
> * **Simplified form of the non-contrastive loss.** We thank the reviewer for pointing out the problem of the non-contrastive loss having a trivial minimizer.
>  We studied the non-contrastive setting as a by-product of our analysis of contrastive losses. During revision, we have put more focus on the contrastive setting and moved the study on non-contrastive setting later to Section 5.
>  This being said, while it is indeed true that,  for arbitrary nonlinear embedding functions $u(x)$, our non-contrastive loss has trivial solution, this cannot happen for deep linear models (studied in Section 5). In the linear setting, the only constant function is the zero function which would not be optimal. Therefore we believe that the analysis of this loss is still interesting. In Section 5, we further note the limitation due to possible trivial solutions.
>
> * **Proposition 3 - Optimization problems e.g. 2\&4 are not equivalent.** Indeed instead of stating 'Then the following optimisation problems are equivalent' Prop 3 should state 'There exists a $W_1, W_2$ that is optimal for all following optimization problems'. We updated Proposition 3 in the current version accordingly.
>  \item Proposition 3 - one negative eigenvalue. We agree that if $C$ only has $1$ strictly negative eigenvalue, then $W_1,W_2$ only need to be of rank $1$ however in this case dimension collapse would also not be a problem. We further note, after Theorem 3, that for a simple Gaussian noise augmentation, $\mathbb{E}[C]$ is negative definite and therefore $C$ would have more then $1$ strictly negative eigenvalue.
>
> * **What does Conjecture 1 add beyond the intuition from NTK?** The flavour of Conjecture 1 is indeed along the lines of known NTK results. However, (i) there is no prior work that proves NTK limit holds for training under (non)contrastive losses (we only know constancy of NTK for squared loss with a known target prediction); and (ii) we are not aware of any statement on NTK in the context optimization under orthogonality constraints.
>
> * **Takeaway from Sec 3.2 / Fig 2.** The observation that increasing the layer width makes the linear and non-linear function closer to each other aligns with the known NTK / lazy training viewpoint. However existing results are derived under a squared loss and in a supervised setting. Sec 3.2 / Fig 2 illustrates that the behaviour can also be observed in the SSL setting under loss function (1) and under orthogonality constraints. Updates in the revised version: We added a clarification together with the previous point to section 3.2.

---

> > ### Author Response · Authors · 2024-04-08
> >
> > ### Related work
> >
> > We thank the reviewer for pointing us to the additional related work and discuss it in the following.
> >
> > * With regards of the theoretical results [1] analyzes the difference in the representation learned by supervised learning and SSL through characterizing the noise component in a planted data model and shows the importance of data augmentation. In contrast our comparison of the supervised and SSL setting through dynamics focuses on the importance of regularization and dimension collapse.
> >  [2] analyzes the phenomena of 'dimension collapse' through the characterization of the loss landscape.
> >  While we also consider the problem of dimension collapse however from the perspective of regularization and weight constraints.
> >  While both works add interesting theoretical insights into SSL we aim to provide a more general viewpoint on SSL through the lens of learning dynamics, which have been shown to be a useful tool to understand the loss landscape and convergence, early stopping, linearised (kernel) approximations and, mostly importantly, generalisation and inductive biases in the supervised setting. We added both references as related theoretical work to the introduction.
> > * The 'Barlow twins loss' [3] naturally restricts the weights from blowing up, and therefore avoids learning trivial representations. This is done by measuring the cross-correlation matrix between the embeddings of two identical networks fed with augmented versions of the data, and aims to make the cross-correlation matrix close to the identity. In comparison we consider a simplified version of losses such as 'SimCLR' [6] or 'MoCo' [7]. Those models are not in a squared loss form and therefore require additional regularization of the embedding function (which we do by orthogonal constraints on the weights) to avoid dimension collapse. We added the reference together with a short discussion to the related works section of the revised version.
> > * We added [4] as prior work on the trace formulation and [5] as prior work for viewing contrastive and non-contrastive objectives in a unifying framework to the revised version.
> >
> >
> > ### Implications of the presented analysis to practically considered, deep SSL models.
> > We discuss this aspect in more detail in the common response posted above to address this common point between reviewers. More specifically we outline how the considered simplifications connect to practically loss functions, how our analysis builds the foundation for analyzing deep non-linear networks and what the potential implications of deriving exact learning dynamics could be in the future.
> >
> > ----------
> >
> > [1] Wen and Li. 21: Toward Understanding the Feature Learning Process of Self-supervised Contrastive Learning.
> > [2] Pokle et al. 22: Contrasting the landscape of contrastive and non-contrastive learning.
> > [3] Zbontar et al. 21: Barlow Twins: Self-Supervised Learning via Redundancy Reduction.
> > [4] Balestriero et al. 22: A Cookbook of Self-Supervised Learning.
> > [5] Garrido et al. 22: On the duality between contrastive and non-contrastive self-supervised learning.
> > [6] Chen, et al. 20: A simple framework for contrastive learning of visual representations.
> > [7] He, et al. 20: Momentum contrast for unsupervised visual representation learning.

---

### Review · Reviewer_nAXY · 2024-02-26

**Summary Of Contributions:**

The paper studies the dynamics of SSL in a novel way. The authors first demonstrate a dimensional collapse issue in SSL without orthogonal regularization. Then they focus on a reformulation of the SSL loss a trace minimization problem with a suitable data-dependent symmetric matrix in the trace operator. The authors show that under certain assumptions (and one conjecture) that the non-linear and linear dynamics settings are close to each other. Then they continue studying the linear setting. The authors accurately understand the dynamics through the eigenvectors of the data-dependent matrix. Numerical evidence is given for the conjectures.

**Audience:**

Yes

**Claims And Evidence:**

Yes

**Requested Changes:**

Please, take a look at the Weaknesses discussion above, and propose changes that can alleviate my concerns. Thanks.

**Strengths And Weaknesses:**

Strengths:

I appreciate studying the problem through the trace minimization formulation. I like the connection the authors make with previous work on the dynamics of the eigenvectors of the data-dependent matrix. I think the work is a fresh perspective of the problem, and it will be useful to practitioners.

Weaknesses:

Some of the motivations of the paper need to come earlier in the text. For example, there is not a concrete definition of dimensional collapse. At many points in the work, there are introductions of some objectives/ dynamics without motivated dynamics. E.g., the trace formulation appears before the authors introduce it later in the text; it is not clear how Lemma 1 connects to Section 2.2 (i.e. what do you mean by SSL not having labels, implies $y=0$, and then how exactly that implies dimension collapse is unclear from the main text); there is not enough intuition presented why Frobenius norm doesn't work for avoiding collapse, while the L2-operator norm works.

I think the work lacks a conclusion that could discuss the impact of the work on SSL as a whole. Currently, there is only a mention of regularization in NTK as an open problem. We need more practical guidance, and high-level insights about the dynamics of SSL in the conclusion.

nit: there are also a few minor typos, e.g. "atleast" <- "at least"; one sumation starts from "i=n" and goes to "n"; and a few others in the text.

---

> ### Author Response · Authors · 2024-04-08
>
> We thank the reviewer for the constructive feedback. We address the concerns below.
>
>
> ### Clarifications on the outlined concerns
>
>
> *  **Introduction of the trace formulation.** We moved the introduction of the trace formulation to earlier in the paper and state it now in the beginning of section 3 in the revised version to make the connection between the considered loss and the trace formulation clearer.
>
> * **Connection between Lemma 1 and Section 2 and SSL not having labels.** Lemma 1 shows that in the supervised setting there is no interaction across output dimensions however this simply implies that the output $u_1,\dots,u_z$ can evolve towards the labels $y\in\mathbb{R}^z$ and therefore each component moves towards a meaningful target. On the other hand in the SSL setup under the contrastive loss there are no labels that drive the components in different directions and therefore components evolve identically. This leads the output to collapse to one dimension at convergence. This is formalized in Proposition 1. While we show this result for simple contrastive (and non-contrastive losses) Remark 2 and 3 outlines that the same still holds for more complex losses and is therefore a problem even in practically more used models.
>
> * **Intuition on why Frobenius norm does not avoid dimension collapse.** The Frobenius norm is defined as the sum of the norms of each column. Critically it does not have any control over the inner products between the columns and no control over the rank. Now recall from the previous point that the problem in the SSL setup is that the outputs can evolve the same as there are no targets. As such by adding a Frobenius norm constraint there is nothing preventing all the columns from being the same and dimension collapse is not prevented. Intuitively if one were given various denominations of coins and asked to pick $k$ coins to maximize value, without a restriction preventing one from picking the same denomination twice, one would simply pick $k$ coins of maximum value. Updates in the revised version: we added the above intuition in the transition from Proposition 2 and 3.
>
>
> ### Implications of the presented analysis to practically considered, deep SSL models.
> We discuss this aspect in more detail in the common response posted above to address this common point between reviewers. More specifically we outline how the considered simplifications connect to practically loss functions, how our analysis builds the foundation for analyzing non-linear deep networks and what the potential implications of deriving exact learning dynamics could be in the future.

---

### Review · Reviewer_xEdX · 2024-03-26

**Summary Of Contributions:**

The paper studies the learning dynamics of SSL methods, which is an important direction. The paper is well-written and makes valid points under the target settings There are a few points, I would appreciate if the authors can address in the revision.

**Audience:**

Yes

**Claims And Evidence:**

Yes

**Requested Changes:**

1. An in depth discussion of the related reference on SSL learning dynamics.
2. To provide further insight between the settings used in this paper and the deep SSL practice.

**Strengths And Weaknesses:**

Strengths:
1. SSL learning dynamics is an important topic.
2. The authors show that the learning dynamics of naïve SSL setting leads to trivial solution, which provides insights to the dimension collapse.
3. The study is extended to the setting with orthogonal constraint, and the author has provide numerical evidence to support the claims.


Potential Weaknesses:
An important reference [1] on SSL learning dynamics is missing. The authors may want to look into this paper and discuss the connection. [1] Simon et al., On the Stepwise Nature of Self-Supervised Learning

Proposition 1 shows that SSL learning objective leads to trivial solution without regularization. And, then the rest of the paper is established upon the orthogonality constraints. While this is a reasonable setting, the results from this paper still has a large gap to the practice in deep SSL. Could you author provide more connection to the deep SSL methods and show how the results in this paper can (significantly) help us in the practice of deep SSL. I would really appreciate if a deep insight can be provided.

---

> ### Author Response · Authors · 2024-04-08
>
> We thank the reviewer for the constructive feedback. We address the concerns below.
>
> ### Additional References.
> In comparison to [1], there are two main differences in our analysis.
> First, [1] derives the learning dynamics of a linear model - embedding function $u(x)= Wx$ - and extend some results to to a kernelised setting $u(x)= W\phi(x)$. In contrast, our main objective is to study a deep model (with more than one trainable layer). In Section 4, we derive dynamics of deep linear model, while the closeness of non-linear models to linear model is stated for 2 layers $u(x)= W_2\phi(W_1x)$.
> Second, [1] considers a simplified version of the 'Barlow twins loss' [2] that minimises the Frobenius norm distance of the covariance/correlation matrix of embeddings from identity matrix. This naturally restricts the weights from blowing up, and therefore avoids learning trivial representations. In contrast, we consider a simplified version of losses such as 'SimCLR' [3] or 'MoCo' [4]. Those models are not in a squared loss form and therefore require additional regularization/restrictions of the embedding function to avoid dimension collapse.
> We incorporate this through the orthogonality constraints on the weights $W_i^{\top} W_i=\mathbb{I}$. Note that orthogonality constraints have been previously introduced for studying the optimisation objective in self-supervised learning [5].
> Updates in the revised version: we added a reference to [1] to the related work section on page 2, noting the difference in considered loss and analyzed embedding function.
>
> ### Implications of the presented analysis to practically considered, deep SSL models.
> We discuss this aspect in more detail in the common response posted above to address this common point between reviewers. More specifically we outline how the considered simplifications connect to practically loss functions, how our analysis builds the foundation for analyzing deep non-linear networks and what the potential implications of deriving exact learning dynamics could be in the future.
>
> ----------
>
> [1] Simon et al., On the Stepwise Nature of Self-Supervised Learning.
> [2] Zbontar et al. 21: Barlow Twins: Self-Supervised Learning via Redundancy Reduction.
> [3] Chen, et al. 20: A simple framework for contrastive learning of visual representations.
> [4] He, et al. 20: Momentum contrast for unsupervised visual representation learning.
> [5] Munkhoeva and Oseledets, 23: Neural Harmonics: Bridging Spectral Embedding and Matrix Completion in Self-Supervised Learning

---

### Author Response · Authors · 2024-04-08
**Implications of the presented analysis on practice of deep SSL**

In this common response we like to address the concern of the reviewers on the practical relevance of the analyzed models and the implications of deriving exact learning dynamics for SSL models. To do so, in the following we first discuss the main model assumptions - namely the considered loss functions as well as the two layer setting. Subsequently, we outline the potential uses of exact learning dynamics.

## Simplifications of the loss function.
  While we consider a simplified form of losses commonly used in practice Remark 2 \& 3 discuss that the simple form still capture the main ideas of such models, and the considered simplifications allow for a theoretical analysis.

## Dynamics beyond 2-layer networks.
  We have now revised the paper and show that the learning dynamics and the convergence of the dynamics hold for arbitrary deep linear networks, which makes the considered model significantly close to ones used in real applications. Figure 6 supports this the new results for deep linear networks as well.
  To further put the obtained results into perspective of existing learning dynamics analysis in the supervised setting let us outline some of the main assumptions taken there.

  Analyzing deep, non-linear networks presents a complex challenge and no unifying approach has been established even in the more studied supervised setting. Existing works usually consider one (or several) of the following assumptions:


* *(a) linear networks.* This allows for an exact characterization of the network [1,2,3] however the considered proof techniques do not extend to the non-linear setting.
* *(b) strong data or initialization assumptions.* [4,5] are able to derive exact solutions in the non-linear setting but the proof structure breaks down if the assumptions (which might not be used in practice) are violated.
* *(c) strong architecture assumptions.* [6,7] derive dynamics for deep, non-linear networks, however need strong assumptions on the initialization and the (infinite) width of the network. How exactly the behaviour of finite and infinite wide networks relate is still an open question.

In this work we aim to analyze settings that are relevant in practice but still allow for a exact theoretical analysis. We address the challenge of analyzing non-linearities and depth by first showing that linear and non-linear networks are close in Theorem 1 and secondly by deriving dynamics for deep linear networks in Theorem 2. In addition our results only have mild initialisation (orthogonality) and data assumptions.

  Updates in the revised version: we updated section 4 so that it is now stated for deep linear networks. Furthermore we added an additional discussion on the challenges of modeling deep non-linear models in the introduction.

## Potential uses of exact learning dynamics.
  The main focus of the presented work is to derive the learning dynamics as they provide a simple and tractable quantity to analyze the network. As outlined in the introduction, in the supervised setting learning dynamics have become an essential tool to understand the loss landscape and convergence [8,1,9], early stopping [10], linearised (kernel) approximations [11,12] and, mostly importantly, generalisation and inductive biases [13,14,15]. Similarly the above quantities are important to study SSL models as well and in this paper we start to build a theoretical foundation to analyze those aspects for the SSL setting.

----------


[1] Saxe et al. 14: Exact solutions to the nonlinear dynamics of learning in deep linear neural networks.
[2] Ziyin et al. 22: Exact solutions of a deep linear network.
[3] Basu et al. 19: Layer dynamics of linearised neural nets.
[4] Tachet et al. 18: On the learning dynamics of deep neural networks.
[5] Mei et al. 22: The generalization error of random features regression: Precise asymptotics and the double descent curve.
[6] Jacot et al. 18: Neural tangent kernel: Convergence and generalization in neural networks.
[7] Arora et al. 19: On exact computation with an infinitely wide neural net.
[8] Fukumizu. 98: Dynamics of batch learning in multilayer neural networks.
[9] Pretorius et al. 18: Learning dynamics of linear denoising autoencoders.
[10] Li et al. 21: Implicit sparse regularization: The impact of depth and early stopping.
[11] Jacot et al. 18: Neural tangent kernel: Convergence and generalization in neural networks.
[12] Du et al. 19: Graph neural tangent kernel: Fusing graph neural networks with graph kernels.
[13] Soudry et al. 18: The implicit bias of gradient descent on separable data.
[14] Luo et al. 19: Towards understanding regularization in batch normalization.
[15] Heckel et al. 21: Early stopping in deep networks: Double descent and how to eliminate it.

---

### Decision · Action_Editor_2oa7 · 2024-06-13

**Recommendation:** Accept as is

**Comment:**

This paper analyzes the training dynamics of self-supervised models, deriving exact learning dynamics under a simplified form of the loss function and with orthogonality constraints on the weights. While the reviewers are in agreement that understanding the dynamics of self-supervised learning is an important question, there was some discussion as to whether the simplifications made here faithfully preserve the important aspects of the problem, especially because there were doubts as to whether the empirical analysis fully closed this gap. On the other hand, the paper clearly describes its limited approach in analyzing simplified settings as a step towards building understanding of more practical configurations. Since the paper does provide sound and convincing evidence supporting this more limited analysis, it seems to fall well within the criteria for publication at TMLR. I recommend acceptance.

**Audience:**

Yes, some members of the TMLR community will be interested in this paper.

**Claims And Evidence:**

Yes, the claims made in the submission supported by the evidence.